# Xylose Isomerase Depletion Enhances Virulence of *Xanthomonas citri* subsp. *citri* in *Citrus aurantifolia*

**DOI:** 10.3390/ijms241411491

**Published:** 2023-07-15

**Authors:** André Vessoni Alexandrino, Evandro Luis Prieto, Nicole Castro Silva Nicolela, Tamiris Garcia da Silva Marin, Talita Alves dos Santos, João Pedro Maia de Oliveira da Silva, Anderson Ferreira da Cunha, Franklin Behlau, Maria Teresa Marques Novo-Mansur

**Affiliations:** 1Laboratório de Bioquímica e Biologia Molecular Aplicada–LBBMA, Departamento de Genética e Evolução, Universidade Federal de São Carlos, São Carlos 13565-905, SP, Brazil; avalex@uol.com.br (A.V.A.); evandrolprieto@gmail.com (E.L.P.); nic.nicolela@gmail.com (N.C.S.N.); 2Programa de Pós-Graduação em Biotecnologia–PPGBiotec, Universidade Federal de São Carlos, São Carlos 13565-905, SP, Brazil; 3Programa de Pós-Graduação em Genética Evolutiva e Biologia Molecular–PPGGEv, Universidade Federal de São Carlos, São Carlos 13565-905, SP, Brazil; joaopedro.maia7@gmail.com (J.P.M.d.O.d.S.); anderf@ufscar.br (A.F.d.C.); 4Fundo de Defesa da Citricultura–Fundecitrus, Araraquara 14807-040, SP, Brazil; tamiris.silva@fundecitrus.com.br (T.G.d.S.M.); talita.santos@fundecitrus.com.br (T.A.d.S.); franklin.behlau@fundecitrus.com.br (F.B.); 5Laboratório de Bioquímica e Genética Aplicada–LBGA, Departamento de Genética e Evolução, Universidade Federal de São Carlos, São Carlos 13565-905, SP, Brazil

**Keywords:** xylose isomerase, pathogenicity, *xylA*, *hrp* genes, *xylR*, citrus canker

## Abstract

Citrus canker, caused by the bacterium *Xanthomonas citri* (Xcc), is one of the most devastating diseases for the citrus industry. Xylose is a constituent of the cell wall of plants, and the ability of Xcc to use this carbohydrate may play a role in virulence. Xcc has two genes codifying for xylose isomerase (XI), a bifunctional enzyme that interconverts D-xylose into D-xylulose and D-glucose into D-fructose. The aim of this work was to investigate the functional role of the two putative XI ORFs, XAC1776 (*xylA1*) and XAC4225 (*xylA2*), in Xcc pathogenicity. XI-coding genes of Xcc were deleted, and the single mutants (XccΔxylA1 or XccΔxylA2) or the double mutant (XccΔxylA1ΔxylA2) remained viable. The deletion of one or both XI genes (*xylA1* and/or *xylA2*) increased the aggressiveness of the mutants, causing disease symptoms. RT-qPCR analysis of wild strain and *xylA* deletion mutants grown in vivo and in vitro revealed that the highest expression level of *hrpX* and *xylR* was observed in vivo for the double mutant. The results indicate that XI depletion increases the expression of the *hrp* regulatory genes in Xcc. We concluded that the intracellular accumulation of xylose enhances Xcc virulence.

## 1. Introduction

Citrus canker is one of the most economically significant citrus diseases, as it affects most of the commercially important citrus species, including oranges, lemons, and limes [1]. The spread of the disease is facilitated by the use of contaminated seedlings and agricultural equipment, by the wind which carries the causative bacteria, *Xanthomonas citri* subsp. *citri* (Xcc), in water droplets, and by the lesions caused by the citrus leafminer (*Phyllocnistis citrella*) [2]. The main symptoms of the disease are canker-like eruptions on leaves, stems, and fruits, which are caused by the hyperplasia of the tissue, rupture of the epidermis, and accumulation of dead cells [1,3]. Although the pathogen is not lethal to the citrus trees, the disease causes a reduction in fruit quality, which restricts its sale in the fresh fruit market [4]. Furthermore, when the fruit has large lesions caused by Xcc, excessive ethylene production is triggered which causes its premature drop, reducing yield and fruit quality [5].

The economic importance of citrus canker control for the citrus industry worldwide and the lack of eco-friendly measures for its control have motivated molecular studies on *Xanthomonas* using omics and functional approaches, such as comparative genomics [6], differential proteomics [7,8,9,10], gene expression analysis [11], genetic analysis through gene deletion [12,13,14], and analysis of protein structure and function [14,15,16]. These studies have expanded the knowledge about the infectious process of Xcc.

The ability of Xcc to use organic sources such as xylose, available during host invasion, may play a role in virulence. Using an in vitro approach, a differential proteomic analysis (2D-PAGE combined with mass spectrometry) was performed for the periplasmic-enriched fractions of Xcc and *X. fuscans* subsp. *aurantifolii* type B (Xca), the causative agent of cancrosis, a milder form of citrus canker [10]. For this, proteins were obtained from both bacteria grown in a pathogenicity-inducing medium (XAM-M), and the enzyme xylose isomerase (XI) was one of the 12 proteins identified from the 4 unique spots from Xcc. Immunodetection also showed that XI expression was preponderant for Xcc relative to Xca under pathogenicity induction. In addition to the involvement of XI, this comparative approach between Xcc and Xca aimed to understand other molecular aspects underlying the virulence of Xcc [10].

Xylose is produced by breaking down xylan, which is the main component of the hemicellulose present in the plant cell wall [17]. *Xanthomonas*, unlike yeasts, is able to utilize xylose, one of the main components of the lignocellulosic mass, which is commonly neglected in industrial biofuel production [18]. The utilization of this abundant and inexpensive biomass by *Xanthomonas* has been the subject of studies examining both biofuel and xanthan gum production [19,20,21]. When xylose is used as the single source of energy by *X. oryzae*, genes from the *hypersensitive response and pathogenicity* (*hrp*) cluster are upregulated [22]. Components of this cluster encode for the Type III Secretion System (T3SS), which plays a role in virulence by allowing the injection of effector proteins directly into the plant cell cytoplasm. 

In bacteria, xylose metabolism involves a series of genes that are often organized into operons and are responsive to this five-carbon sugar. In *Bacillus* spp. and *Lactobacillus pentosus*, the *xylAB* operon includes the *xylB* (xylulokinase, XK) and *xylA* (xylose isomerase, XI) genes, which are downregulated by XylR, a regulator of the *Lac*I family [23,24]. In *Caulobacter crescentus*, XylR binds to and dissociates from the *xyl*-box operator in the absence and presence of xylose, respectively, allowing the transcription of the genes related to the xylose metabolism when this carbohydrate is present [25]. In *X. oryzae*, XylR represses the expression of various *xyl*-box-regulated genes involved in xylose metabolism, as well as *hrp* genes [26]. The deletion of the *xylR* gene in *Streptomyces lividans* increased the expression of the XI gene, one of the key enzymes of xylose metabolism, almost five-fold [27]. 

According to Zandonadi and colleagues, the two putative XI genes (*xylA*) found in the annotated genome of Xcc at NCBI (XAC1776 and XAC4225, referred to herein as *xylA1* and *xylA2*, respectively) are 1338 bp in length, and their predicted amino acid sequences share 99.3% identity. Despite this similarity, the genomic contexts of the two genes are not the same, and only *xylA2* is preceded by a predicted *xyl-*box, suggesting that *xylA2* may be regulated by XylR. On the other hand, the genome of Xca in NCBI contains only one complete ORF encoding for XI (XAUB_09030), which shares 97% identity with Xcc ORFs and is also preceded by a predicted *xyl*-box [10]. 

XI (EC. 5.3.1.5), also known as glucose isomerase, is an enzyme responsible for catalyzing the reversible reaction of isomerization of D-xylose to D-xylulose and of D-glucose to D-fructose [28]. XI has an affinity for xylose, which is 160 times more abundant than glucose in *Aeromonas hydrophila* [29]. D-xylose units are linked by β (1 → 4) linkage to form xylan, a heteropolysaccharide that is one of the major plant cell wall hemicelluloses. In Xcc, D-xylose assimilation involves the transport of D-xylose across the cytoplasmic membrane, isomerization to D-xylulose by XI, and phosphorylation by XK to D-xylulose-5-phosphate, a compound that is ultimately metabolized via the pentose phosphate pathway [17].

Although the metabolism of xylose has been hypothetically linked to the pathogenicity of Xcc, there have been no functional studies investigating the role of XI in the infectious process of this bacterium. In this study, we have (i) confirmed the genomic annotation of the biological function of Xcc *xylA*, (ii) demonstrated a differential role of *xylA1* and *xylA2* genes in the interaction between Xcc and *Citrus aurantifolia*, and (iii) provided evidence that xylose metabolism is involved in citrus canker pathogenesis.

## 2. Results

### 2.1. XylA2 Has Isomerase Activity for Glucose and Xylose

The yield of the purified recombinant xylose isomerase (rXylA2) was 5 mg.L^−1^ of *E. coli* culture. Purification was performed by one-step Ni-affinity chromatography due to the recombinant protein to be fused to a His tag, encoded by the pET plasmid. Figure 1 shows that rXylA2 (~52 kDa) was primarily expressed in the soluble fraction of the lysate and was most efficiently eluted and purified between imidazole concentrations of 125 and 500 mM. The enzymatic activities of rXylA2 were first confirmed for the conversion of D-xylose to D-xylulose by observing the decrease in color intensity of the reaction product as rXylA2 concentration decreased (Appendix A). The activity for the conversion of D-fructose to D-glucose was found to increase as a function of both time and rXylA2 concentration (Appendix A).

### 2.2. Confirmation of xylA Deletion in the Mutants 

Gene deletions were confirmed by colony PCR using oligonucleotides that hybridize with Xcc genomic regions adjacent to the flanking regions of the target genes (Figure 2). For all reactions using DNA from wild-type colonies as a template, the PCR product had an expected size of approximately 3.4 kb, which corresponds to the sum of 1 kb from the upstream region, 1338 kb from the coding region of the *xylA1* or *xylA2* genes, and 1 kb from the downstream region (Figure 2, lanes 1, 5, 9, and 13). For all reactions that used DNA from mutant colonies (XccΔxylA1, Xcc∆xylA2, and XccΔxylA1∆xylA2), the PCR product was ~2 kb in size, corresponding to the sum of each 1 kb of the upstream and downstream regions (Figure 2, lanes 2, 6, 10, and 14).

Colony PCR analysis confirmed the deletions of the regions of interest in XccΔxylA1, XccΔxylA2, and XccΔxylA1∆xylA2. The PCR products had the expected size for just the *xylA* flanking regions (*xylA* absence) and were susceptible to digestion with *Bam*HI (Figure 2, lanes 3, 7, 11, and 15), a restriction site inserted after double homologous recombination. The results confirmed that the PCR products corresponded to each deleted region and that the oligonucleotides did not hybridize in any of the deletion vectors (pNPTS_xylA1 and pNPTS_xylA2) as no bands were amplified from PCR reactions using such plasmid vectors as template DNA (Figure 2, lanes 4, 8, 12, and 16).

### 2.3. Xcc and Deletion Mutants Grow Differentially in the Presence and Absence of Xylose

The growth of Xcc and the three *xylA* deletion mutants were assessed in the absence or presence of xylose (10 mM), as shown in Figure 3. Under xylose absence, the growth of Xcc was higher than that observed for XccΔxylA1ΔxylA2, but similar to XccΔxylA1 and XccΔxylA2. This may indicate that XI ORFs, XAC4225, and XAC1776, present in each of the mutants, or even both in Xcc, have a role in supporting bacterial growth, independently of the presence of external xylose. When xylose was added, there was no decrease in the growth of Xcc, XccΔxylA1, and XccΔxylA2 at ~50 h, as observed when xylose was absent (Figure 3), suggesting that the presence of both or even one XI ORF may contribute to a more efficient utilization of extracellular xylose. The double mutant showed a longer lag phase, both in the presence and absence of xylose, when compared to the wild-type strain and the two single mutants. However, regardless of the presence of xylose, all deletion mutants for *xylA* were able to grow on XAM-M, a nutrient-poor and pathogenicity-inducing medium. 

*Tukey* analysis was performed, and growth did not show significant differences between the strains when grown in XAM-M. However, on XAM-X, Xcc growth was greater than that of XccΔxylA1 (*p* = 0.0012) and XccΔxylA2 (*p* = 0.0019). Growth did not show significant differences between XccΔxylA1, XccΔxylA2, and XccΔxylA1ΔxylA2 strains grown in XAM-X. The growth of the *Xcc* (*p* ≤ 0.0001), XccΔxylA1 (*p* = 0.0007), XccΔxylA2 (*p* = 0.0002), and XccΔxylA1ΔxylA2 (*p* = 0.0001) strains increased when xylose was added (Figure 3a,b).

### 2.4. xylA Deletion Increases Virulence of Xcc in C. aurantifolia

The mutant strains along with the wild type were evaluated for pathogenicity in *C. aurantifolia*, and the intensity of the symptoms in leaves infiltrated with the different assessed strains was visually different (Figure 4). As expected, infiltration with the saline (0.9% NaCl) solution did not result in the development of symptoms (Figure 4a). The wild strain of Xcc (Figure 4b) produced expected citrus canker symptoms when infiltrated with high inoculum concentration, including soaking, chlorosis, development of pustules (mainly on the abaxial side), and some necrosis. In leaves infiltrated with XccΔxylA1 (Figure 4c), XccΔxylA2 (Figure 4d), and XccΔxylA1ΔxylA2 (Figure 4e), the symptoms were more severe and produced a larger necrotic area when compared to the wild type (Figure 4b). Furthermore, symptoms produced by XccΔxylA2 (Figure 4d) and XccΔxylA1ΔxylA2 (Figure 4e) were notably more intense than those caused by XccΔxylA1 (Figure 4c) and the wild strain (Figure 4b). All three mutant strains produced larger necrotic areas in infiltrated leaves than Xcc. The statistics and the necrotic areas show differences, with the percentage of the necrotic area being 8% for *Xcc*, 41.8% for XccΔxylA1, 55% for XccΔxylA2, and 66.4% for XccΔxylA1ΔxylA2 (Figure 4f).

### 2.5. xylA Deletion Affects the Expression of xylR and hrp Regulators Mainly in the Double Mutant

Xcc and mutant strains expressed *xylR*, *hrpG*, and *hrpX* during in vitro growth in the XAM-M medium, irrespective of xylose addition or growth phase, and in vivo (Figure 5a–c). For the in vitro growth, there was no difference in the expression of *xylR* in XAM-M (OD_595nm_ = 0.4 and 0.8) and in XAM-X (OD_595nm_ = 0.4). However, for the late phase of the in vitro culture (OD_595nm_ = 0.8) and with the addition of xylose (XAM-X), the expression of *xylR* was higher in XccΔxylA1ΔxylA2 when compared to Xcc (*p* = 0.0049), XccΔxylA1 (*p* = 0.0102), and XccΔxylA2 (*p* = 0.0015). In vivo, the same *xylR* expression pattern was observed, with *p*-values equal to 0.0001, 0.0001, and <0.0001 for Xcc, XccΔxylA1, and XccΔxylA2, respectively (Figure 5a). 

The *hrpG* expression did not show significant differences between the strains grown in XAM-M and XAM-X at OD_595nm_ 0.4. However, when grown in XAM-M at an optical density of 0.8, the expression of *hrpG* in XccΔxylA1 was higher than in Xcc (*p* = 0.0017) and XccΔxylA2 (*p* = 0.0005). Under this growth condition, XccΔxylA1ΔxylA2 showed even higher *hrpG* expression compared to Xcc (*p* < 0.0001) and XccΔxylA2 (*p* < 0.0001). In XAM-X at OD_595nm_ 0.8, *hrpG* expression was higher in Xcc compared to XccΔxylA1 (*p* = 0.0119), XccΔxylA2 (*p* = 0.0020), and Xcc ΔxylA1ΔxylA2 (*p* = 0.0266). In vivo, XccΔxylA1 exhibited a higher expression of *hrpG* compared to Xcc (*p* = 0.0005) and XccΔxylA2 (*p* = 0.0009), as well as XccΔxylA1ΔxylA2 compared to Xcc (*p* = 0.0027) and XccΔxylA2 (*p* = 0.0067) (Figure 5b). 

In vivo, *hrpX* gene expression was higher in XccΔxylA1ΔxylA2 compared to Xcc (*p* = 0.0009), as well as the XccΔxylA1 (*p* = 0.013) and XccΔxylA2 (*p* = 0.0002) strains. In all in vitro cultures, the evaluated strains showed the same level of *hrpX* expression.

## 3. Discussion

The infection process of *Xanthomonas* spp. involves a combination of events, including bacterial adhesion [30], adaptation and survival in the stress conditions of the phyllosphere [31], colonization of the apoplast [32], and degradation of the plant cell wall [17]. In the latter process, bacteria employ cell wall degradation enzymes (CWDEs) to break down the components of the plant cell wall, which acts as a physical barrier against bacterial invasion [33]. These enzymes specifically target hemicellulose, the major component of the plant cell wall, through xylanolytic systems that include xylanases, β-xylosidases, arabinofuranosidases, acetyl xylan esterases, and α-glucuronidases [17,34]. 

In recent years, the role of enzymes involved in the degradation and utilization of xylan and xyloglucans has been extensively studied in various species of *Xanthomonas*, revealing that these enzymatic systems play a crucial role in the pathogenicity and virulence of xanthomonads towards plants [35,36]. 

Following the release of xylose monomers by xylanolytic systems, the enzymes XI and XK convert xylose into xylulose and xylulose-5-phosphate, respectively, enabling integration with the pentose phosphate pathway [34]. The Xcc genome contains two predicted ORFs for XI, noted as XAC1776 (*xylA1*) and XAC4225 (*xylA2*). While their amino acid sequences are 99% identical, the genomic and regulatory contexts of these two genes are vastly different, primarily because *xylA1* appears to be part of an operon, whereas *xylA2* is likely transcribed from a monocistronic mRNA unit under the regulation of a putative regulatory region named *xyl*-box (TGGTAGCGCTAACA) [10,34]. The *xylA1* operon has been shown to be involved in xyloglucan depolymerization and encompasses various glycosidic hydrolases, a modular xyloglucan acetyl esterase, and specific membrane transporters. Sugars released by this system trigger the expression of several virulence factors, including the Type III Secretion System (T3SS) [36].

Previous studies have demonstrated that the *xyl*-box of *xylA* functions as a regulatory region to which the XylR repressor protein binds in *X. campestris* pv. *campestris* [34], *X. oryzae* pv. *oryzae* [26], *C. crescentus* CB15 [25,37], and *Lactobacillus pentosus* [24]. In *X. campestris* pv. *campestris* and *X. oryzae* pv. *oryzae*, the presence of xylose inactivates XylR as a repressor of transcription from *xyl*-box-regulated genes, which encode enzymes involved in xylan degradation and xylose utilization. Additionally, the interaction between XylR and xylose in *X. oryzae* pv. *oryzae* results in the accumulation of HrpX, a positive regulator of the *hrp* [26,34] and the T3SS genes in *X. fuscans* subsp. *fuscans* [31]. Furthermore, when Xcc is grown in a xylose-amended medium, the T3SS genes are upregulated [36]. 

In this study, we characterized the function of the two genes coding for XI (XAC1776 and XAC4225) in Xcc. We obtained Xcc deletion mutants for each one of the two XI copies. Lack of XI complementation for the deletion mutants could be considered a shortcoming of this work. However, deletion was performed not only for one gene, but independently for two XI genes, which share 99.3% identity for the predicted amino acid sequence and are located at distinct regions of the Xcc genome. Importantly, not only each single mutant, but also the double mutant presented increased virulence. These consistently altered phenotypes in the mutants in relation to the wild strain which would be highly unlikely to occur by chance. These findings indicate that xylose accumulation caused by XI depletion may interfere with the pathogenesis of citrus canker by exacerbating the virulence of Xcc. 

The two predicted enzymatic activities for Xcc XylA were confirmed experimentally for the rXylA2. These activities were therefore considered not to be performed by the deletion mutants when both *xylA1* and *xylA2* deletions were carried out or at least diminished when only one of the genes was deleted. Although only the recombinant XylA2 was experimentally assayed in this study, these similar activities can be inferred for XylA1, which shares 99% identity with XylA2. 

Xcc cells lacking XylA activity displayed slower initial growth in in vitro culture in plant-mimicking medium, but exhibited intensified virulence in vivo, suggesting that xylose not metabolized for energy conservation and/or carbon assimilation by the cell can act as a signal for cell preparation for host invasion. This indicates that xylose plays a role beyond being a nutrient for Xcc. From this perspective, the isomerization of this carbohydrate to xylulose may also play a role in attenuating virulence in Xcc. In other words, XI could be a negative virulence factor in the wild-type strain, in which both *xylA* genes are present, whereas the absence of one or two *xylA* genes may lead the mutants to be more virulent. Furthermore, when *xylA2* remained intact (as in XccΔxylA1), disease symptoms were attenuated compared to XccΔxylA1ΔxylA2 (Figure 5).

Our results indicate that the regulatory mechanism of *hrp* genes and, consequently, bacterial virulence, are affected by xylose accumulation in the *xylA* mutants. Because there are no predicted alternative metabolic pathways for xylose utilization, the deletion of the XI-coding genes is expected to lead to the intracellular accumulation of xylose during the Xcc infectious process. This may allow the excess xylose to bind the XylR regulatory protein and result in a lower affinity of XylR to the *xyl*-box, and, consequently, increased expression of the genes involved in xylan degradation (or xylose utilization) and *hrp* regulator genes expression. In *X. oryzae* pv. *oryzae*, the quantitative regulation of *hrp* genes via HrpX is dependent on the sugar source, as the presence of xylose in the cultures promotes the accumulation of this protein, which does not occur with other sugars such as glucose, fructose, and sucrose. Additionally, it has been demonstrated that the proteolysis of HrpX is an important *hrp* regulatory mechanism [22], as the interaction between xylose and XylR can derepress genes encoding protease inhibitors, decreasing the proteolysis of HrpX and inducing the expression of *hrp* genes [36].

The results revealed that XccΔxylA1—and, notably, XccΔxylA2 and XccΔxylA1ΔxylA2—were more virulent than Xcc. In this case, upon the deletion of *xylA1*, xylose utilization may become dependent on the XI expressed by *xylA2* under XylR regulation. The higher virulence of XccΔxylA2 and XccΔxylA1ΔxylA2 suggests that *xylA2*, the XI copy probably regulated by XylR, is a determinant for conversion of xylose to xylulose, particularly when xylose is largely available, as in the case of advanced host wall degradation. Thus, *xylA2* deletion, more than *xylA1* deletion, seems to contribute to a higher accumulation of xylose that could bind to XylR and decrease HrpX proteolysis, resulting in the induction of *hrp* regulator gene expression and the increased virulence of mutant strains.

These findings indicate that *xylA2* expression in the wild strain may play a prominent role in virulence attenuation, as its expression is modulated by xylose accumulation itself, which also triggers the expression of genes required for host cell invasion. Thus, the transitory repression of XylA2 expression (due to XylR repression) or, alternatively, the saturation of the XylA2′s ability to metabolize all available xylose are possibilities that could lead to intracellular xylose abundance. This carbohydrate could act as an intracellular signal for the presence of an external host cell wall partially degraded by the xylanolytic system and, consequently, more susceptible to the action of the T3SS apparatus. In this sense, interrupting this signaling can reduce the virulence of Xcc, which could be evaluated in the future by constructing strains that overexpress XI. 

Although further confirmation that XylA2 has a predominant catalytic role over XylA1 is required, the increased virulence caused by the deletion of *xylA2* (XccΔxylA2 and XccΔxylA1ΔxylA2) compared to XccΔxylA1 and Xcc after 20 days of infection suggests, at least in this phase of in vivo growth, a differential role of the two XI-encoding genes in the pathogenesis of citrus canker. In corroboration, Xca, the causative agent of a milder disease in citrus (cancrosis), harbors only one complete XI ORF (97% identical to the two Xcc ORFs), apparently under *xyl*-box control [10]. This reinforces that *xylA2*, a gene under the *xyl-box* control, has a major effect in the alleviation of bacterial virulence, helping maintain controlled levels of intracellular xylose.

The RT-qPCR analysis of *xylR*, *hrpG*, and *hrpX* expression revealed that after an initial period of in vivo growth (about 5 days of host infection), these regulator genes had, compared to wild Xcc, higher expression, particularity in XccΔxylA1ΔxylA2, a double mutant which is expected to accumulate more intracellular xylose (Figure 5). However, at this early phase, XccΔxylA1 also presented higher expression of *hrpG* in relation to XccΔxylA2 (Figure 5). This is somewhat unexpected, as in planta experiments have shown that *xylA2* has a prominent catalytic activity over *xylA1*, as *xylA2* deletion produced a more virulent mutant. The unexpected difference in the level of *hrpG* transcripts between XccΔxylA1 and XccΔxylA2 is likely related to the fact that the cellular uptake of xylose from the host, after only 5 days of infection, is not sufficiently high to derepress the *xylA2* expression (*xyl-box* regulated) in the XccΔxylA1 mutant. As a result, an increase in xylose concentration (and, consequently, *hrpG* induction) was possible, whereas *xylA1* could be expressed freely (since it is not XylR-regulated) in the XccΔxylA2 mutant and was able to prevent an excess of xylose accumulation. Although in vivo experiments showed that the XccΔxylA2 mutant was more virulent, this was observed at a later phase of infection (20 days). At this stage of in vivo growth, xylose intracellular levels can attain higher levels due to the extensive degradation of the host wall, and *xylA1* expression is insufficient to metabolize all available xylose. 

The RT-qPCR analysis of xylR expression in Xcc did not show any significant changes during in vitro growth, even in the presence of xylose (Figure 5). This is in line with previous studies, as Ikawa and colleagues did not find differences in the *xylR* expression of *X. oryzae* pv. *oryzae* when grown in a medium containing xylose or glucose [26]. Dejeán and collaborators did not detect changes in the *xylR* expression of *X. campestris* in the presence of xylose [34]. Blast searches indicate that, like Xcc, genomes available at the NCBI of *X. campestris* and *X. oryzae* strains have two copies of XI. Since XylR has a regulatory role and its function is inactivated by xylose binding, it seems reasonable that XylR would not be reset through compensatory synthesis, as this would negate the modulatory effect of xylose on the regulation of gene expression. 

The results of our study suggest that in the Xcc double mutant, which lacks the ability to convert xylose to xylulose, increased *xylR* expression not only in vivo but also at the late phase of bacterial growth in XAM-X, a poor-nutrient medium containing xylose (Figure 5). In such situations, it is expected that xylose attains abnormally high intracellular levels compared to Xcc, especially because the double mutant is unable to convert xylose to xylulose. This suggests that XI could work as an intracellular xylose scavenging system, and the deactivation of this system, more than simply supplying external xylose, is necessary to increase XylR expression in *Xanthomonas*. It is also possible that the interruption of the xylulose flux to the pentose phosphate pathway in the double mutant could be another influential factor, but further research is needed to confirm this. 

Our findings indicate that the presence of the two genes encoding XI in Xcc may provide advantages over Xca, which does not have an xylR-independent xylA copy [10], potentially allowing Xcc to utilize xylose at lower concentrations as an energy and carbon source.

## 4. Materials and Methods

### 4.1. Bacterial Strains, Culture Media, Conditions, and General Procedures 

Wild Xcc strain 306 (IBSBF 1594) was supplied by the Fundo de Defesa da Citricultura (Fundecitrus, Araraquara, SP, Brazil) and stored at −80 °C in Luria–Bertani (Sigma; St. Louis, MO, USA) medium with 10% glycerol. The Wizard Genomic DNA Purification Kit (Promega; Madison, WI, USA) was used to extract Xcc genomic DNA. Phusion High-Fidelity DNA Polymerase, restriction enzymes *Nde*I, *Hind*III, *Bam*HI, *Nhe*I, and *Xho*I, pJET 1.2 cloning vector, and other DNA extraction kits were purchased from Thermo Fisher Scientific (Waltham, MA, USA). Deletion vector pNPTS138 (Alley, unpublished results) was provided by Prof. Dr. Henrique Ferreira (UNESP, Rio Claro, SP, Brazil). The ampicillin and kanamycin antibiotics were purchased from Sigma (St. Louis, MO, USA). All other reagents were of analytical purity. General molecular biology procedures were performed as previously reported [38] or as described.

### 4.2. XI Heterologous Expression for Confirmation of the Two Predicted Enzymatic Activities 

The *xylA2* (ORF XAC4225) codifies for XylA2 that has a predicted amino acid sequence 99% identical to XylA1 (ORF XAC1776), evidencing that either one of the two ORFs could be cloned to assess XI activity in Xcc. XI coding sequence (ORF XAC4225) was amplified by PCR using genomic DNA from Xcc as template and primers specifically designed to result in an amplification product containing the *Nde*I and *Xho*I restriction sites (Table 1). PCR amplification was performed on a C1000 Touch Thermal Cycler (Bio-Rad, Hércules, CA, USA), with 50 μL total volume reaction containing 500 ng of Xcc genomic DNA, 2 μM from each primer, 2.5 U of Phusion High-Fidelity DNA Polymerase, 1.5 mM MgCl_2_, and 0.8 mM dNTP (0.2 mM each). After amplification at 98 °C for 10 min, 30 cycles at 98 °C for 30 s, 60 °C for 30 s, and 72 °C for 1.5 min, and a final extension step at 72 °C for 10 min, PCR product was gel purified, cloned into the pJET 1.2 vector, and used in *E. coli* DH5α transformation. Ampicillin-resistant recombinant clones were confirmed by restriction analysis and nucleotide sequencing [39] at Genoma-USP (https://genoma.ib.usp.br/en, São Paulo, SP, Brazil). The excised insert was cloned in the *Nde*I- and *Xho*I-digested pET28a expression vector (Novagen, Rahway, NJ, USA), which was used in *E. coli* DH5α transformation. The plasmid pET28a_xylA2, obtained from a kanamycin-resistant clone, was used in the transformation of *E. coli* BL21(DE3) and an isolated colony was grown in LB broth containing kanamycin 30 µg.mL^−1^ until OD_595nm_ = 0.4. Following this, 0.1 mM IPTG was added to the culture, which was incubated in an orbital shaker at 250 rpm for 19 h at 18 °C. Cells were collected by centrifugation (12,000× *g*, 20 min at 4 °C), suspended in 50 mM Tris-HCl pH 8 buffer and 100 mM NaCl, and sonicated (Sonic Dismembrator 500, Fisher Scientific, Hampton, NH, USA) under ice bath. The lysate was then centrifuged for 10 min at 4 °C and 12,000× *g* to separate the soluble and insoluble cellular fractions. Purification of the recombinant XI (rXylA2) was performed by immobilized metal affinity chromatography (IMAC) in a 5 mL Ni-NTA column (Novagen, Rahway, NJ, USA) using the soluble fraction of the cell lysate. Flow-through and fractions eluted by an imidazole concentration gradient (7 mM to 500 mM) were analyzed by SDS-PAGE [40], together with the soluble and insoluble cellular fractions. Recombinant protein, eluted in 500 mM imidazole, was dialyzed against 50 mM Tris-HCl (pH 8.0) and 100 mM NaCl, quantified by the UV absorption method at 280 nm and utilized in the enzymatic activity analysis.

The two predicted activities of XI were assayed for the purified recombinant protein. The first predicted activity of XI is the isomerization of D-xylose into D-xylulose, and the Seliwanoff test was used to detect the formation of ketose in a reaction containing the recombinant XI (rXylA2), D-xylose, and Seliwanoff reagent [41]. Three assays were prepared with the following concentrations of purified recombinant enzyme: 0.05, 0.15, and 0.25 mg.mL^−1^. Each assay received 20 µL of 1% resorcinol, 600 µL of 30% HCl, 300 µL of 0.5 M of xylose, and distilled water to a total volume of 1.4 mL. Two controls were prepared: the blank received 50 mM Tris-HCl (pH 8.0) and 100 mM NaCl in replacement of the enzyme and xylose solution volumes, maintaining all proportions of the other compounds. The positive control consisted of 0.5 M fructose instead of xylose, and no enzyme. First, rXylA2 and its substrate were incubated for 3 h. Thereafter, HCl with resorcinol and distilled water was added, solutions were vigorously shaken and incubated at 80°C for 30 min, and color change was monitored. The second predicted activity of XI is the interconversion of D-glucose into D-fructose. To verify this activity, a test based on the coupled reactions of glucose oxidase and peroxidase was employed to detect glucose in the solution [42]. For this, fructose was used as substrate and the Glucose Liquiform kit (Labtest, Lagoa Santa, MG, Brazil) was used for detection, which is composed of 30 mM phosphate buffer pH 7.5, 1 mM phenol, glucose oxidase 12,500 U/L, peroxidase 800 U/L, 290 μM 4-aminoantipyrine, 7.5 mM sodium azide, and surfactants. Three different concentrations of rXylA2 were used: 20, 60, and 100 μg.mL^−1^. Glucose production was measured by analyzing, in triplicate, the absorbance at 490 nm after 2 and 3 h of reaction at room temperature, using a previously constructed standard curve (glucose at 0.1–1 mg.mL^−1^).

### 4.3. Plasmid Construction for Deletion of xylA1 or/and xylA2 Genes from Xcc

Mutant strains were obtained using the vector pNPTS138, following a methodology standardized by our research group [12,14,15]. For the construction of each deletion plasmid, 1 kb regions flanking the *xylA1* (ORF XAC1776) and *xylA2* (ORF XAC4225) genes, obtained from the Xcc genome sequence at NCBI, were PCR amplified using Xcc genomic DNA as template and *xylA up* or *xylA down* oligonucleotide pairs, shown in Table 1 for upstream or downstream flanking regions, respectively. PCR was initiated with a denaturation step at 98 °C for 10 min, followed by 30 cycles at 98 °C for 30 s, 64 °C for 30 s, and 72 °C for 1 min, and by a final extension step at 72 °C for 10 min. PCR products were cloned into pJET 1.2 vector and sequenced [39]. For each deletion vector, the flanking regions for each *xyl*A gene were excised from the pJET 1.2 vector and cloned in tandem into the pNPTS138 vector. For this, the pNPTS138 vector was digested with enzymes compatible with one of the flanking regions’ ends, fragment ligated, and transformed in *E. coli* DH5α. The recombinant plasmid obtained was then digested with the enzymes compatible with the second flanking region, and ligation and *E. coli* transformation were performed to obtain each deletion plasmid [12,14]. 

### 4.4. Construction of Deletion Mutants Lacking One or Both Xcc xylA Genes 

Each deletion vector was used for Xcc transformation by electroporation [43], followed by plating on LB agar containing kanamycin at 30 μg.mL^−1^. The vector pNPTS138 carries a copy of the gene coding for the enzyme levansucrase (*sacB*, from *Bacillus subtilis*), which converts sucrose into a cell-toxic compound [44]. Antibiotic-resistant colonies that were unable to grow on a sucrose-amended medium are expected to have the plasmid integrated in the genome by a first homologous recombination. These selected colonies were repeatedly cultured overnight in the absence of kanamycin in LB broth to allow the second recombination event for plasmidial excision and curing. The culture was then plated on LB agar containing 10% sucrose. Only the bacteria that had the plasmid excised from the genome (together with the target gene) were able to grow on the sucrose-containing medium. Cells harboring a single recombination were also eliminated by having the plasmid integrated into the genome. 

Xcc mutants for each *xylA* gene (XccΔxylA1 and XccΔxylA2) were confirmed by PCR using oligonucleotide pairs that hybridize with the region located 50 bp apart from the flanking fragments (Table 1). In addition, after the confirmation of the deletion of ORF XAC4225, a single mutant strain was transformed again with the deletion vector containing the flanking regions of ORF XAC1776 to obtain a double mutant strain (XccΔxylA1ΔxylA2), following the procedures described above.

### 4.5. Growth Curves for Xcc and xylA Deletion Mutants in the Presence and Absence of Xylose

Growth curves of Xcc, XccΔxylA1, XccΔxylA2, and XccΔxylA1ΔxylA2 strains were performed in XAM-M pathogenicity-inducing medium composed of 7.57 mM (NH_4_)_2_SO_4_, 33.06 mM KH_2_PO_4_, 60.28 mM K_2_HPO_4_, 1.7 mM citrate sodium, 1 mM MgSO_4_, 0.03% casamino acids, 10 mM fructose, 10 mM sucrose, and 1 mg.mL^−1^ BSA, pH 5.4 [8], in the absence or presence of 10 mM xylose (XAM-X). Cultures of 400 mL were carried out at 28 °C and 200 rpm and OD_595nm_ was monitored every 24 h. The growth patterns of the curves were analyzed by calculating the area under the curves, followed by the *Tukey* multiple comparison test, with a confidence level of 95%.

### 4.6. In vivo Pathogenicity Assays of Xcc and xylA Deletion Mutants

Wild Xcc and XccΔxylA1, XccΔxylA2, and XccΔxylA1ΔxylA2 deletion mutants were tested for pathogenicity in potted *C. aurantifolia* trees in a greenhouse. This host, known as ‘Mexican lime’, is highly susceptible to citrus canker [1], and is therefore suitable for assessing the pathogenicity of mutant strains. Isolated colonies of these strains were cultured in 5 mL of LB broth until reaching OD_595nm_ = 0.4 (5 × 10^8^ CFU.mL^−1^). Then, 100 μL of the bacterial suspensions was centrifuged at 12,000× *g* for 15 min at 28 °C and the pellets were suspended in 9.9 mL of 0.9% saline, resulting in a suspension containing 5 × 10^6^ CFU.mL^−1^. Approximately 150 μL of the suspensions or the saline solution, used as a negative control, was infiltrated individually into the abaxial side of the leaves with a 5 mL needleless syringe. Four experimental replicas (four leaves) were performed for each treatment. All treatments were tested in each one of the plants to minimize the biological variation in the host defense response in the results. Twenty days post-infiltration (dpi), leaves were detached from plants and photographically recorded. The assay was repeated using biological replication (an independent culture). The images of the leaves were analyzed with ImageJ software, and the calculation of necrotic areas involved the use of the arcsine square root transformation. The statistical analysis was conducted using Duncan’s test, with a confidence level of 95%.

### 4.7. RT-qPCR Analysis for xylR and hrp Regulator Gene Expression in Xcc and Deletion Mutants

The expression of *hrpG*, *hrpX* [22], and *xylR* [17] regulator genes was assessed in Xcc and XccΔxylA1, XccΔxylA2, and XccΔxylA1ΔxylA2 deletion mutants. The *xylR* gene is annotated at NCBI for Xcc 306 genome as *salR*. Firstly, gene expression was analyzed under in vitro bacterial growth in XAM-M and XAM-X medium. Cultures were kept under shaking at 200 rpm until OD_595nm_ = 0.4 (early phase) and 0.8 (late phase), following the centrifugation of 5 mL in an Eppendorf microcentrifuge (12,000× *g*, 5 min). For in vivo experiments, bacterial suspensions were inoculated into *C. aurantifolia* in a greenhouse as previously described. After 5 days, two 1 cm diameter disks were excised from infiltrated areas and macerated in a lysis buffer from the Total RNA Purification Kit (Cellco Biotec do Brasil Ltda; São Carlos, SP, Brazil) used for in vitro and in vivo RNA extractions following the manufacturer´s instructions.

cDNA was synthesized from a 5 µg RNA aliquot previously treated with DNAseI (Invitrogen^TM^, Waltham, MA, USA), using 10 pmol of each of the reverse primers for the regulatory genes *xylR*, *hrpG*, and *hrpX* (*xylR*-R, *hrpG*-R, and *hrpX*-R, respectively, Table 2), 10 mM dNTPs, and 200 U Script Reverse Transcriptase (Cellco Biotec do Brasil Ltda, São Carlos, SP, Brazil) in RT Buffer (250 mM Tris-HCl pH 8.3, 500 mM KCl, 30 mM MgCl_2_, and 25 mM DTT). The reactions were performed in a C1000 Touch Thermocycler (Bio-Rad, Hércules, CA, USAfor 10 min at 25 °C, 90 min at 37 °C, and 15 min at 70 °C. The *atpD* and *gyrB* genes were used as reference controls (reverse primers *atpD*-R730 and *gyrB*-R, respectively, Table 2). As the results were similar for both reference genes, we presented only *atpD* results [45]. Primers for RT-qPCR were designed using OligoAnalyzer 3.1 (Integrated DNA Technologies, Coralville, IA, USA) and are listed in Table 2.

The melting curve was determined by varying the temperature from 95 °C for 15 s, 60 °C for 1 min, and 95 °C for 15 s. The relative fold change in mRNA quantity was calculated according to the 2^(−ΔΔCt)^ method, and all values were normalized to the expression of the *atp*D (XAC3649) and *gyr*B (XAC3896) genes [46].

## 5. Conclusions

XI participates in the xylose metabolism and is a negative virulence factor for the pathogenesis of citrus canker, not being essential for the bacterial growth. A greater influx of xylose into Xcc due to the interaction with the citrus host requires that both XI copies are being expressed to efficiently scavenge intracellular xylose; however, as long as repression of the second XI copy by XylR persists, albeit temporarily, there will be intracellular accumulation of xylose and the consequent induction of the *hrp* genes. This suggests that xylose, internalized into Xcc after release from the host cell wall by the action of the bacterial xylanolytic system, may constitute an intracellular signal that triggers T3SS expression to enable host infection by Xcc.

## Figures and Tables

**Figure 1 ijms-24-11491-f001:**
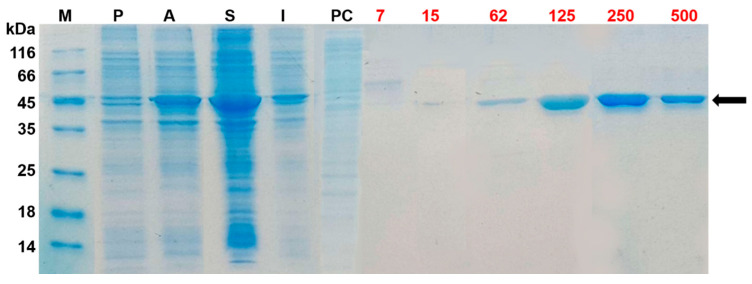
SDS-PAGE analysis of the heterologous expression of rXylA2 and purification by Ni-affinity chromatography. M: Protein molecular weight marker (Thermo Fisher Scientific, Waltham, MA, USA); P: Cellular lysate prior to IPTG induction of gene expression; A: Cellular lysate after IPTG induction of gene expression; S: Soluble fraction of the cellular lysate; I: Insoluble fraction of the cellular lysate; PC: Soluble fraction of the lysate after its passing through the immobilized nickel column (post-column). The protein elution was performed by applying a gradient of imidazole concentration (whose concentrations in mM are shown in red). The band of rXylA2 overexpression is indicated by the arrow.

**Figure 2 ijms-24-11491-f002:**
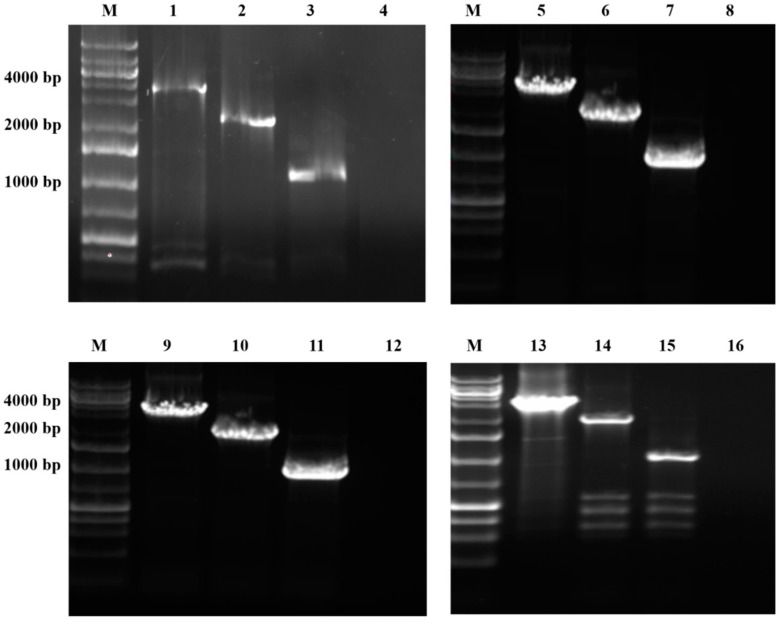
Confirmation of the *xylA* gene deletion in mutants by PCR using oligonucleotides that hybridize to regions outside of *xylA* 1 kb upstream and downstream regions. Analysis of PCR products resulting from reactions using wild and mutant Xcc colonies as templates, as well as the pNPTS138_xylA1 and pNPTS138_xylA2 deletion vectors. M|GeneRuler 1 kb Plus DNA Ladder Molecular Size Standard (Fermentas, Waltham, MA, USA). 1 and 9|PCR products using wild Xcc as template and oligonucleotides for confirmation of *xylA1* deletion. 2|PCR product using XccΔxylA1ΔxylA2 as template and oligonucleotides for confirmation of *xylA1* deletion. 3|PCR product from lane 2 digested with *Bam*HI. 4|PCR product using pNPTS138_xylA1 vector as template. 5 and 13|PCR products using wild Xcc as template and oligonucleotides for confirmation of *xylA2* deletion. 6|PCR product using XccΔxylA1ΔxylA2 as template and oligonucleotides for confirmation of *xylA2* deletion. 7|PCR product from lane 6 digested with *Bam*HI. 8|PCR product using pNPTS138_xylA2 vector as template. 10|PCR product using XccΔxylA1 as template and oligonucleotides for confirmation of *xylA1* deletion. 11|PCR product from lane 10 digested with *Bam*HI. 12|PCR product using pNPTS138_xylA1 vector as template. 14|PCR product using XccΔxylA2 as template and oligonucleotides for confirmation of *xylA2* deletion. 15|PCR product from lane 14 digested with *Bam*HI. 16|PCR product using pNPTS138_xylA2 vector as template. Relevant bands of the molecular size standard are indicated.

**Figure 3 ijms-24-11491-f003:**
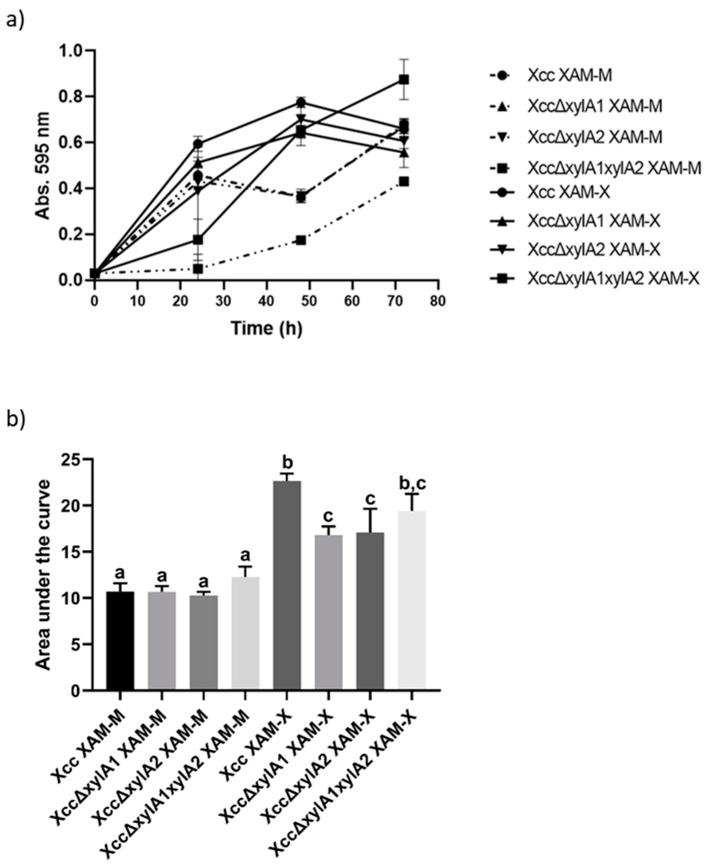
Growth curves and area under the curves. Growth curves of Xcc, XccΔxylA1, XccΔxylA2, and XccΔxylA1ΔxylA2 in XAM-M pathogenicity-inducing medium, in the presence (XAM-X) or absence of xylose (XAM-M). The 400 mL cultures were carried out at 28 °C and 200 rpm and the xylose concentration used was 10 mM (panel (**a**)). Comparison of the area under the growth curves between strains grown with (XAM-X) and without the addition of xylose (XAM-M) to the medium (panel (**b**)). Statistical analysis was performed using the Tukey test (*p* = 0.05). Groups with equal lowercase letters showed means without statistical differences.

**Figure 4 ijms-24-11491-f004:**
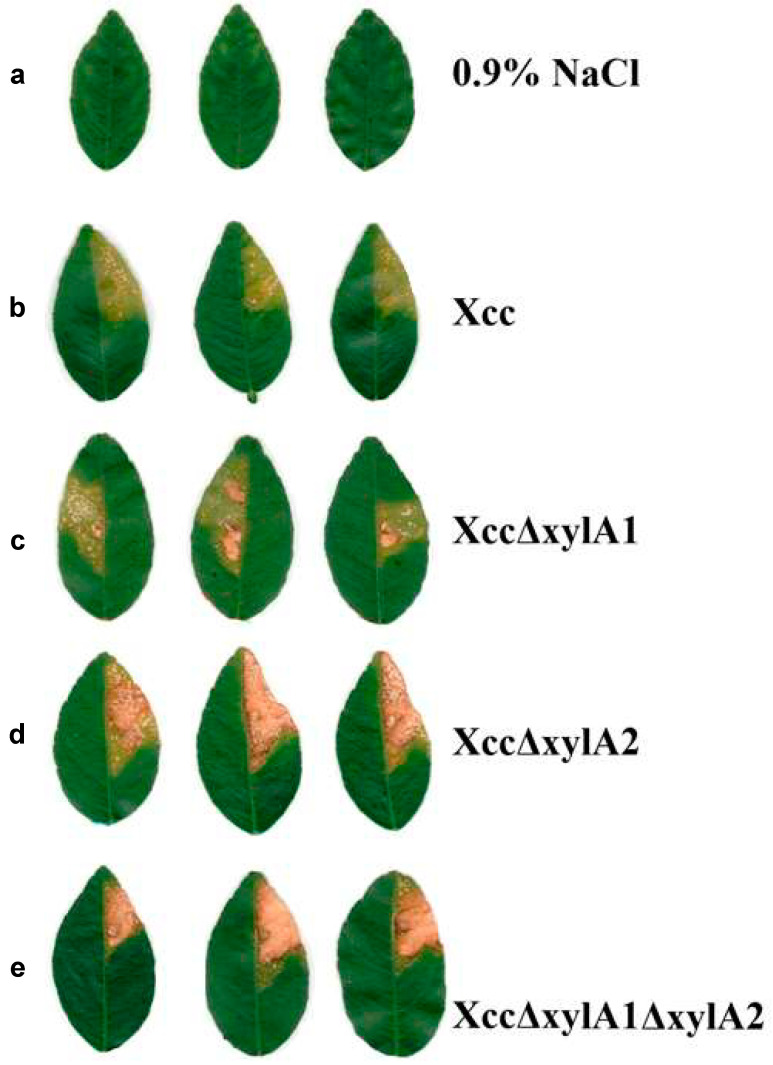
Evaluation of pathogenicity in *C. aurantifolia* using Xcc, XccΔxylA1, XccΔxylA2, and XccΔxylA1ΔxylA2 deletion mutants. Six leaves were infiltrated with each strain, and three representative ones are shown in the figure. A negative control using saline solution (0.9% NaCl) was also performed. The images shown in (**a**–**e**) were captured on the twentieth day post-infiltration. The necrotic areas (**f**) were calculated using the arcsine square root transformation. Statistical analysis was performed using Duncan’s test (*p* = 0.05). Groups with equal lowercase letters showed means without statistical differences.

**Figure 5 ijms-24-11491-f005:**
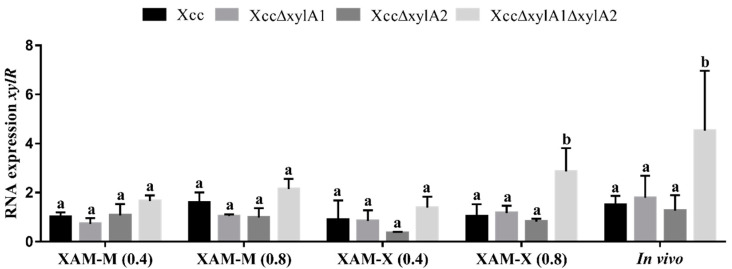
Analysis of the expression of *xylR*, *hrpG*, and *hrpX* genes in Xcc and in the deletion mutants. Bacterial cultivation was carried out in vitro using XAM-M medium (without xylose) and XAM-X medium (XAM-M with 10 mM xylose) at an optical density of 0.4 and 0.8. In vivo growth was performed on *C. aurantifolia* leaves, and the analysis was performed using cells obtained from 1 cm-diameter discs after 5 days of infection. The results were represented by values of 2^−ΔΔCt^ normalized relatively to the control *atpD*. Error bars indicate the standard error of the mean. Groups with equal lowercase letters showed means without statistical differences.

**Table 1 ijms-24-11491-t001:** Primers utilized for PCR analysis in Xcc and deletion mutants. Amplifications were performed for Xcc *xylA2* gene, up or down flanking regions of *xylA1* and *xylA2*, and deletion region (up plus down). Underlined sequences in the nucleotide sequences indicate the restriction sites.

Gene or Region/Utilization	Primer Nucleotide Sequence	Product Size (bp)	Restriction Enzymes for Sites in Respective F and R Primers
ORF XAC4225 (*xylA2* gene)/heterologous expression	F:TATATACATATGAGCAACACCGTGTACAT R: CTCGAGTCAACGCGTCAGATACTG	1338	*Nde*I*/Xho*I
*xylA1* up/ deletion vector	F: TATATAAAGCTTGGCTGGACGTGCGC R: TATTAGGATCCGGGGTGTGAAGTCCTTG	1000	*Hind*III*/Bam*HI
*xylA1* down/ deletion vector	F: TATATAGGATCCTGTCCCGTGGCCGG R: TATATAGCTAGCCTGCTCGGAGAAGCCC	1000	*Bam*HI/*Nhe*I
*xylA2* up/ deletion vector	F: TATAAAGCTTGGTCACGCCATGCGTC R: TTAATAGGATCCGGGGTGAAGCTCCTG	1000	*Hind*III/*Bam*HI
*xylA2* down/ deletion vector	F: TATATATAGGATCCGCCTTGCCACTGCAC R: TATATAGTCGACGGTGGCATCGCGTAC	1000	*Bam*HI/*Sal*I
up + down of XccΔxylA1/ deletion confirmation	F: GCTCACCGGCGAGCGCTT R: AGGCACCGATGCTGAGGCC	2000 (mutant) ~3300 (wild)	-
up + down of XccΔxylA2/ deletion confirmation	F: GATGGTGGCCGAGCGCGAT R: GCTGCTGGGCGTGTTGCG	2000 (mutant) ~3300 (wild)	-

**Table 2 ijms-24-11491-t002:** Primers for RT-qPCR analysis of *hrpG*, *hrpX*, and *xylR* expression in Xcc and mutants.

Gene Name	Primer Nucleotide Sequence (RT-qPCR)	Size (bp)	Efficiency (%)	Concentration (mM)
*atpD* (control, XAC3649)	F: CGGCGCACCGTCGTAT R: CCGGTTTCCAGCAATTCG	53	106.096	100/100
*gyrB* (control, XAC3896)	F: CGTCCCGGCATGTATATCG R: ACCACCTCGAACACCATGTGA	67	102.371	100/100
*hrpG* (XAC1265)	F: CAGCACATCTACAAGTTGCG R: CCTTGCTCATTGTCGTTGC	100	100.868	100/100
*hrpX* (XAC1266)	F: CGATGATGAGGTCAGTTTGT R: ACTGCGCAAAGCAATTCAAC	100	99.297	100/100
*xylR* (XAC4226)	F546: AGCCAAAGAGATCACCGAAC R730: GGCCGGATTTGTAGGTGTAA	166	91.96	100/100

## Data Availability

Not applicable.

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
