# Peer review of "Xylose Isomerase Depletion Enhances Virulence of Xanthomonas citri subsp. citri in Citrus aurantifolia"

_ijms, 2023, doi:10.3390/ijms241411491_

Round 1

Reviewer 1 Report (Previous Reviewer 2)

"Lack of complementation do not invalidate the results presented, which are all unreported in literature" is not acceptable at all. Present quality science specially in molecular biology need complementation for a quality publication. if your double mutant produced 100% lesions, this outcome could have been speculated. How do you explain any epigentic issues without complementation? In your discussion, mention that  lack of complementation is a major shortcoming in your paper. Please plan ahead to include complementation data in your future papers.

Minor grammar check up is required.

Author Response

Please see the attachment files of the manuscript and other documents

Reviewer 2 Report (Previous Reviewer 1)

Some letters should be italicted. For example, “p=0.0012” should be “p = 0.0012”

Please check whether “p=<0.0001” is right. Sometime the “P” is capitalized in the text.

The  title of Figure 3 should be rewrote.

Author Response

This manuscript is a resubmission of an earlier submission. The following is a list of the peer review reports and author responses from that submission.

Round 1

Reviewer 1 Report

Xcc is the pathogen of citrus canker. Xcc has two genes codifying for cylose isomerases. This study focuse on the relationship of these two genes to the xylose and their results to the pathogenicity. They concluded that xylose accumulation as a result of XI depletion may enhance Xcc virulence.

Major:

For the growth curve assay, no replications of the experiment was performed? I think it should be repeated at least 3 times. So that in figure 3 and 2.3 part, the higher or lower growth of the bacteriums is significant or not could be present with p values.

In the In vivo pathogenicity assays, can xylose be applied to the C. aurantifolia. Leaves? I think the authors should add this additional experiment.

Minor:

It seems the authors are not very sure about the conclusions. For example, in the abstract, they used the word “mayfor 3 times in Lines 18, 26, and 28. Even in the conclusion, they wrote “suggesting that xylose accumulation may trigger exacerbation of virulence in Xcc”.

Lines 32-34, Unlike citrus Huanglongbing, the other most important bacterial disease, the citrus canker severity on different citrus cultivars are quite different.

Lines 35-36, some insects with chewing mouthparts. e.g. Leaf miner, are also play important roles in the disease transmission.

rXylA2 is short for recombinant XI. But when it first appeared in Line 108, no full name was provided. Similarly for C. aurantifolia. Please check others.

Line 129, ml-1 should be mL-1

 Why do you select C. aurantifolia as material? Is it highly susceptable to Xcc? The background of this cultivar should be introduced in the M&M.

Reviewer 2 Report

This paper needs major modifications  as suggested below. This could be a strong paper once these modifications are done.  

Line 129- Confirmation of xylA deletions in the mutants: PCR is not a reliable method to confirm gene deletion and southern blot with correct size bands are recommended/required to confirm gene deletions.

Line 183 and Fig.3. Statistics need to be applied if you claim growth curves are different from treatment to another.

Line 204 and Fig.4- need to collect more leaves data and  lesions areas of individual leaves and apply statistics to show the differences if you claim c vs d &e are different. Present data in a table and show statistics.

Finally, please complement the gene (insert the gene back) and show it revert back to WT level in terms of pathogenicity.

Round 2

Reviewer 1 Report

This manuscript is much better in this edition. Overall, the design of the research is straigtforward and easy to follow.  There is no experimental flaw and the data is sufficient for publication with minor revision. Please check the manuscript throughly.

Reviewer 2 Report

Lack of Complementation study, leaves  lesion comparative statistics and growth curve comparison are not acceptable.